# Correcting Model misspecification via Generative Adversarial Networks

## Abstract

Machine learning models are often misspecified in the likelihood, which leads to a lack of robustness in the predictions. In this paper, we introduce a framework for correcting likelihood misspecifications in several paradigm agnostic noisy prior models and test the model's ability to remove the misspecification. The "ABC-GAN" framework introduced is a novel generative modeling paradigm, which combines Generative Adversarial Networks (GANs) and Approximate Bayesian Computation (ABC). This new paradigm assists the existing GANs by incorporating any subjective knowledge available about the modeling process via ABC, as a regularizer, resulting in a partially interpretable model that operates well under low data regimes. At the same time, unlike any Bayesian analysis, the explicit knowledge need not be perfect, since the generator in the GAN can be made arbitrarily complex. ABC-GAN eliminates the need for summary statistics and distance metrics as the discriminator implicitly learns them, and enables simultaneous specification of multiple generative models. The model misspecification is simulated in our experiments by introducing noise of various biases and variances. The correction term is learnt via the ABC-GAN, with skip connections, referred to as skipGAN. The strength of the skip connection indicates the amount of correction needed or how misspecified the prior model is. Based on a simple experimental setup, we show that the ABC-GAN models not only correct the misspecification of the prior, but also perform as well as or better than the respective priors under noisier conditions. In this proposal, we show that ABC-GANs get the best of both worlds.

**Keywords:** Likelihood-free inference, Deep Neural Regression, Approximate Bayesian Computation, GAN

## 1 Introduction

A model is a probing device used to explain a phenomenon through data. In most cases, a true model for this phenomenon exists but cannot be specified at all [Le & Clarke (2017)]. This setting indicates that all plausible models, though useful, can be deemed as misspecified [Box (1976)]. Can we use a plausible explainable model, while correcting for its misspecification implicitly? Unlike the prescriptive generative modeling dogma, predominant in the statistical community, the implicit generative modeling view taken by the machine learning community lays emphasis on predictive ability rather than on explainability [Brieman (2001)]. Implicit Deep Generative Models have witnessed tremendous success in domains such as Computer Vision. However, their opaqueness and lack of explainability has made the injection of subjective knowledge into them a highly specialized and experimental task. In this work, our proposal is to reconcile implicit and explicit generative models into a single framework in the misspecified setting. We do that by taking GANs and ABC as representative of the two fields respectively.

The introduction of GANs in 2014 by Goodfellow et al. [Goodfellow et al. (2014)] marked a very decisive point in the field of generative deep learning. Since then, deep learning based generative models like GANs and Variational Autoencoders have been extensively worked on, with the main intention of addressing issues with likelihood estimation based methods and related strategies. The crux of these issues lies in complex or intractable computations that arise during maximum likelihood estimation or evaluation of the likelihood function. A GAN uses two adversarial modules - the Generator and the Discriminator, essentially playing

a zero sum min-max game with each other, with the competition between them driving both modules to improve and reach a stage where the Generator is able to produce counterfeit data which is indistinguishable from the real data. Although GANs have been shown to address the issues mentioned above well by leveraging the benefits of using piece-wise linear units, there are some inherent issues with the GAN paradigm. These include the inherent difficulty in achieving convergence, stability issues during training and the necessity of large amounts of data. An active area of research in this direction is to apply GANs in different settings [Karras et al. (2018); Radford et al. (2016)] and also to improve stability [Gulrajani et al. (2017)].

Another older, but equally interesting generative paradigm is Approximate Bayesian Computation (ABC) [Beaumont et al. (2002)] [Beaumont (2010)] [Grelaud et al. (2009)] [Csilléry et al. (2010)] [Marin et al. (2012)]. ABC finds its roots in Bayesian inference, and aims to bypass likelihood evaluation by approximating the posterior distribution of model parameters. This method is extremely useful in cases when the likelihood estimation is computationally intensive, or even intractable. The likelihood-free aspect of this paradigm allows the data generative model to be as complex as it can get. However, there are some drawbacks, such as low acceptance rates at higher dimensions, the difference between the prior distribution from the posterior distribution, identification of lower dimensional statistics to summarize and compare the datasets and the model selection problem.

**ABC and GAN complementarity:** Looking at these two paradigms, it becomes clear that both ABC and GANs try to solve the same problem - learning the data generation mechanism by capturing the distribution of the data, but they approach the problem in different ways. By studying these two paradigms, their similarities and differences become apparent. With respect to the data generation model, ABC uses a user-specified model, whereas the Generator in a GAN is non-parametric. Looking at the discriminative model for both, ABC uses an explicit, user-specified discriminator which often uses Euclidean distance or some other distance measure on a set of summary statistics to measure the difference between real and simulated datasets. For GANs, the discriminative model is specified through a function like KL divergence or JS divergence as the Discriminator's objective function. Another key difference here is that the feedback from the Discriminator in a GAN flows back to the Generator, thereby making them connected, while in ABC, these two modules are disconnected. Further, in ABC, model selection is followed by model inference, but in GANs, since the Generator and Discriminator are connected, this occurs implicitly during the learning process. We now see that ABC and GAN appear to be at two ends of the data generation spectrum, with each having its own advantages and disadvantages.

## 2 Motivation and Contributions

As it is clear from the previous discussion, both GANs and ABCs are likelihood-free methods. But there are certain limitations to both of them. ABC is a Bayesian paradigm. Like in any Bayesian modeling approach, subjective knowledge about the data generating model is expressed both in terms of the likelihood (explicit or implicit) and the prior. One would want to exercise more freedom in the choice of priors, however. Majority of the model selection criteria focus on the priors, keeping the likelihood fixed. However, misspecification in the likehood can lead to spurious errors and make the inference invalid. Some model selection criteria like Deviance Criteria (DIC) won't work well in such cases. It is generally a hard problem to tackle computationally, if one were to obtain marginal evidences. So how we do address this problem? In the context of ABC, the choice of summary statistic and the distance metric to compare the simulated datasets with the real data set determine the efficiency of the approximation. While it seems advantageous to rely on sampling, it leaves many issues suggested above to experimentation and to the modeler. Model selection and sensitivity analysis have to be performed, regardless. Can we get rid of making choices about the summary statistics, distance metrics, model selection in the context of ABC? Further, can we deal with model misapplication either in likelihood or prior or both in the Bayesian context? GANs, in particular, the adversarial mini-max formulation can address these questions.

However, GANs require relatively large amounts of data, owing to their non-parametric nature, to train the Generator and Discriminator networks. It is also known that training GANs can be unstable [Kodali et al. (2017)]. A consequence of deep networks, of which GANs are a special case, is that, they are opaque from the standpoint of interpret ability [Li et al. (2022)]. Further, incorporation of any available prior knowledge into

GANs is limited to modifying the architectures or loss functions or a combination of them. In part, it may be due to the long held misconception that deep learning eliminates the need for good feature engineering. However, good feature engineering gives a way to architecture design. Can we incorporate prior knowledge into GANS? Can the GANs work on low-data regimes, where prior knowledge could be both available and valuable? We argue that, ABC can augment the Generator network of a GAN. The amount of correction needed can be learnt via the data itself, without making hard choices *a priori*.

We show the effectiveness of our work through several ABC-GAN models. We consider cGAN [Mirza & Osindero (2014a)] and TabNet [Arik & Pfister (2019)] as baseline GANs with some architectural modifications.

1. mGAN: GAN Generator takes as inputs the features, and the simulated data from ABC.

2. skipGAN: GAN Generator takes as inputs the features, and the simulated data from ABC, and the Discriminator, also takes a weighed combination of ABC Generator and GAN Generator.

3. Tab-mGAN: mGAN with TabNet as the Generator of the GAN.

4. Tab-skipGAN: skipGAN with TabNet as the Generator of the GAN.

They are described in detail later. We consider several standard, interpretable models such as Linear Models, Gradient Boosted Trees (GBT) and a combination of Deep Learning and Gradient boosted trees (TabNet) as ABC models under various misspecification settings. Extensive experimentation (check sections (4), (5) helps us answer and tackle the questions posted above, and shows the novelty of our work.

## 3 Our Approach

Some notations and settings are introduced to make the exposition clear. Let $\mathcal{D}_\tau = \{y_i^\tau, x_i^\tau\}_{i=1}^n$, be the observed dataset, a set of $n$ i.i.d tuples $(y_i^\tau, x_i^\tau)$, where $x_i^\tau \in \Re^p$ is a p-dimensional column feature vector, and $y_i^\tau \in \Re$ is the response variable. Assume that, $G_\tau$ is the true generative model, typically unknown, that produces $y_i^\tau \sim G_\tau(x_i^\tau)$. Define the datasets $\mathcal{D}_\pi \equiv \{y_i^\pi, x_i^\pi\}_{i=1}^n$ and $\mathcal{D}_\gamma \equiv \{y_i^\gamma, x_i^\gamma\}_{i=1}^n$, that can be sampled by ABC and GAN, respectively. Here, by convention, $y_i^\pi \sim G_\pi(x_i^\tau), x_i^\pi = x_i^\tau$ for ABC and similarly $y_i^\gamma \sim G_\gamma(x_i^\gamma)$. Further, assume that $d(.,.)$ is some distance or loss such as Mean Absolute Error (MAE) that measures discrepancy between two datasets. Note that $G_\pi$ is typically a sampler and $G_\gamma$ is a deterministic transformation.

### 3.1 ABC-GAN Framework

Suppose that we know the generative model $G_\pi$, but it is misspecified. In order to rectify this misspecification, we append it to a standard GAN generator $G_\gamma$ network, i.e., $x_i^\gamma = [y_i^\pi, x_i^\tau]$. $G_\gamma$ now transforms $G_\pi$ samples so as to make resulting dataset look more realistic. Now, by design, $G_\gamma$ can be quite shallow. The hope, rather, intent is that, the "sampler" is already pretty good, and lot of domain knowledge is encoded in it. Therefore, not much needs to be done by the $G_\gamma$, except doing a few corrections. The exact corrections that are to be done are taught by the Discriminator of the GAN $D_\gamma$. Under ideal conditions, when perfect knowledge about the Sampler $G_\pi$ (the pre-generator, or the generative model in the context of ABC) is known, $G_\gamma$ does an identity transformation. Under these conditions, the GAN learning should not be a concern (stability-wise), as the problem is already regularized. From an architecture perspective, $G_\gamma$ can have large capacity but is regularized to produce an identity transformation. Hence, the primary objectives are to investigate two key hypotheses

1. when $G_\pi$ is perfect, i.e, $G_\pi = G_\tau$, we expect $G_\gamma = I(.)$, an identity map and $d(\mathcal{D}_\tau, \mathcal{D}_\pi) = 0$.

2. when $G_\pi$ is imperfect, $G_\gamma \neq I(.)$ and $d(\mathcal{D}_\tau, \mathcal{D}_\pi) > 0$. More than that, we expect $d(\mathcal{D}_\tau, \mathcal{D}_\pi) > d(\mathcal{D}_\tau, \mathcal{D}_\gamma)$.

We consider two broad families of architectures to test the hypothesis.

**mGAN**: We depict the functional architecture of baseline overall model *mGAN*, shown in Fig. 1. The GAN is a vanilla cGAN except that one of the inputs is the ABC Generator's output, i.e., $x_i^\gamma = [y_i^\pi, x_i^\tau]$ and $\mathcal{D}_\gamma \equiv \{y_i^\gamma, x_i^\gamma\}_{i=1}^n$ will be passed to the Discriminator $D_\gamma$.

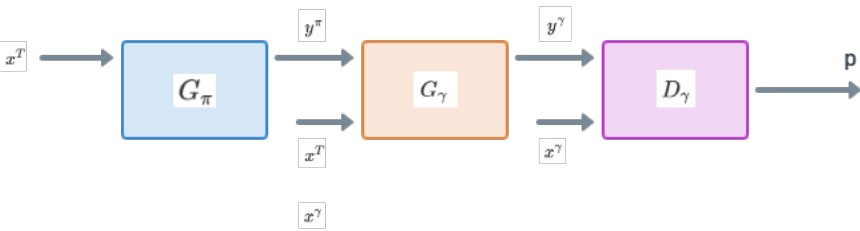

Figure 1: A baseline mGAN model

.

**skipGAN**: Another variant that we experimented with is the *skipGAN*. We conjecture that vanilla mGAN might have information bottleneck. When the prior model is very good, both $G_\gamma$ and $D_\gamma$ can be very shallow. If not explicitly regularized, training mGAN could be hard. We can mitigate this problem by supplying both $y_i^\pi$ and $y_i^\gamma$ to $D_\gamma$. Specifically, we choose weighed average so that the weights can be seen as model averaging, and can also be interpreted as amount of expressiveness borrowed by mGAN. That is, $D_\gamma$ gets $wy_i^\pi + (1-w)y_i^\gamma$. The idea of using skip connection is to try to achieve performance improvement over mGAN. At the least, it should be able to ensure that the mGAN does at least as well as the baseline $G_\pi$.

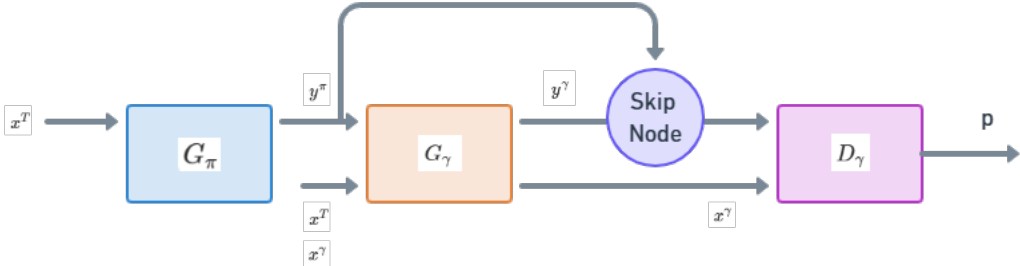

Figure 2: Proposed skipGAN model

### 3.2 Objective Function

Consider the following hybrid generative model:

$$p_i = D_\gamma(G_\gamma([y_i^\pi, x_i^\tau]), \text{ with } y_i^\pi \sim G_\pi(x_i^\tau)$$

Then the likelihood can be written as: $L(y) = \prod_{i=1}^n p_i$. In fact, it is striking to see the likelihood as empirical likelihood [Owen (2001)] without the normalization constraint $\Sigma p_i = 1$. But it is not obvious how to estimate $D_\gamma$ and $G_\gamma$, if not for the adversarial min-max optimization used in GANs. In that sense, we see that our contribution is more in using the adversarial optimization to maximize the empirical likelihood, that has absorbed a non-parametric correction term by Deep Neural Networks and some prior models.

## 4 Experimental Setup

Several experiments were conducted to test the impact of the ABC-GAN models in correcting mis-specified prior models $G_\pi$. The purpose of these experiments is two-folds: 1) to assess the benefits of including a prior for a GAN and 2) to verify that ABC-GAN models successfully correct misspecified models.

We consider three datasets (one simulated and two real), three prior generative models, two basic ABC-GAN architectures, and two GAN Generator architctures - leading to a total 36 experiments. The misspecification at the prior generative model has bias and variance terms with three levels each. Each of the 36 experiment has 9 runs each simulating different amounts of model misspecification, taking the total number of experiments to 324. The details are provided below.

## 4.1 $G_\pi$: Prior Generative Models

In particular, we consider, Linear Models, Gradient Boosting Trees, and Transformers for $G_\pi$ and Feedforward Networks and Transformers for $G_\gamma$. We also consider different Ground Truth generative models $(G_\tau)$. Under perfect information $G_\pi = G_\tau$. For simulated datasets, $G_\tau$ is known. Imperfect information can creep from mis-specified sampling distribution or prior or both. We simulate misspecifcation by adding Gaussian noise to an assumed $G_\tau$. To keep the design space smaller and simpler, we consider mis-specified priors, keeping the likelihood of the prior generative model fixed. We consider three families of models - Linear Models, Gradient Boosted Trees (GBTs), and Transformers - as explicit generative models.

1. Linear Models: Standard Linear Regression models are implemented in statsmodel [Seabold & Perktold (2010)], a Python module that provides classes and functions for the estimation of many different statistical models, as well as for conducting statistical tests, and statistical data exploration. We use the linear ordinary least squares model as our baseline.

2. GBTs: CatBoost [Dorogush et al. (2018)] is an algorithm for gradient boosting on decision trees. It is developed by Yandex researchers and engineers, and is used for search, recommendation systems, personal assistant, self-driving cars, weather prediction and many other tasks. It is an industry standard and an ambitious benchmark to beat. We use *catboost* implementation.

3. Transformers: TabNet, a Transformers-based models for Tabular data, was introduced in [Arik & Pfister (2019)]. This model inputs raw tabular data without any preprocessing and is trained using gradient descent-based optimisation. TabNet uses sequential attention to choose which features to reason from at each decision step, enabling interpretability. Feature selection is instance-wise, e.g. it can be different for each row of the training dataset. TabNet employs a single deep learning architecture for feature selection and reasoning, this is known as soft feature selection. These make the model enable two kinds of interpretability: local interpretability that visualizes the importance of features and how they are combined for a single row, and global interpretability which quantifies the contribution of each feature to the trained model across the dataset. We use TabNet as baseline by calling the TabNetRegressor class under pytorch-tabnet module.

Henceforth, all references to Stats Models, CatBoost, and TabNet, correspond to Linear Models, GBTs and Transformers, respectively, where applicable.

In this other extreme case, we pass covariates $(x_i^\tau)$ plus random noise $(e_i)$ to GAN, i.e., $x_i^\gamma = [x_i^\tau, e_i]$ in which case, the ABC-GAN acts more like a conditional-GAN [Mirza & Osindero (2014b)].

## 4.2 $G_\gamma$: GAN Generators

We consider two architectures:

1. *Feed Forward Networks (FFN):* The FFN Generator consists of 5 hidden layers of 50 nodes each and ReLU activation. The Discriminator consists of 2 hidden layers of 25 and 50 nodes respectively followed by ReLU activation after every layer.

2. *Transformers:* We consider the same TabNet Regressor used in $G_\pi$ discussed earlier- the Transformer-based Generator.

### 4.3 Model Misspecification

The following noise model is considered for real datasets:

$$y_i^\pi \sim N(y_i^\tau + \mu, \sigma^2)$$

For the Linear Model, we consider a full Bayesian model, of the following specification:

$$y_i^\pi \sim N(<x_i^\tau, \beta>, 1)$$

where $\beta \sim N(\beta^\tau + \mu, \sigma^2)$ with $\mu \in \{0, 0.01, 0.1\}$, $\sigma^2 \in \{0.01, 0.1, 1\}$ and $\beta^\tau$ is a true, pre-specified part of the generative model, $y_i^\tau$ is the output of the prior model $G^\pi$ and $<, >$ is the standard dot product.

### 4.4 Datasets

We evaluate our models on the following Synthetic and real datasets:

*1. Friedman3* [Friedman (1991)] consists of 4 independent features $z = [z_1, z_2, z_3, z_4]$ as input, uniformly distributed on the intervals: $0 \le z_1 \le 100$, $40\pi \le z_2 \le 560\pi$, $0 \le z_3 \le 1$, $1 \le z_4 \le 11$. The generative model for $y$ is is nonlinear model $y = \arctan((z_1 z_2 - 1/(z_1 z_3))/z_1)$. A standard normal noise is added for every sample. The dataset has 100 samples.

*2. Boston:* The Boston Housing Dataset [Harrison & Rubinfeld (1978)] is derived from information collected by the U.S. Census Service concerning housing in the area of Boston MA. The dataset has 503 samples and 13 columns/features.

*3. Energy efficiency* [Tsanas & Xifara (2012)] contains eight attributes and two responses (or outcomes. The dataset has 768 samples. The aim is to use the eight features to predict each of the two responses. For our experiments, we have restricted only to the first response with all 8 features.

### 4.5 Training

The cGAN, mGAN, skipGAN and their TabNet versions are trained for 1000 epochs with BCE loss function and a batch size of 32. The dataset is split into training and validation sets (80-20) and the same validation set is used to validate the performance of all models. The learning rate used for Friedman 3 dataset is 0.001, and is 0.01 for all other datasets. All datasets are run using 1.6 GHz Dual-Core Intel Core i5 CPUs.

### 4.6 Metrics

We use MAE to evaluate the performance of the models. The experiments were run 10 times and the average of the MAE over the 10 runs is presented.

## 5 Results

In order to test the hypothesis that, ABC-GAN models perform no worse than the prior models, we take Boston dataset, and synthetically inject model misspecification, as described earlier, and report MAE of $G_\pi$ (sampler) and $G_\gamma$ (a deterministic transformation). In Fig. 3, we show the boxplots of the MAE for each of the models, for each of the prior models. As can be seen, the proposed ABC-GAN models outperform the prior models in almost all cases - different priors, different ABC-GAN models, and different levels of model misspectications. Even a simpler mGAN successfully corrects the misspecified baselines (Linear Models, Boosted trees and TabNet) and results in lower MAEs than the prior model. Next, we investigate, how these models perform at specific levels of model misspecification by prior, model architecture, and dataset.

In Tables 1-9, each row corresponds to a level of model mispecification as indicated by (Variance, Bias) columns, and rows corresponding to columns - Prior Model, mGAN, Tab-mGAN, skipGAN, Tab-skipGAN - indicate the MAE of the models indicated by the column header. In the case of skip variants, the skip

| Variance | Bias | Prior model | mGAN | Tab-mGAN | skipGAN | Weights skipGAN | Tab-skipGAN | Weights Tab-skipGAN |
|---|---|---|---|---|---|---|---|---|
| 1 | 1 | 1.3310 | 0.4918 | 0.3310 | 0.4594 | 0.9932 | **0.3005** | 1.0000 |
| 1 | 0.1 | 1.0820 | 0.5060 | 0.6127 | **0.4245** | 0.6657 | 0.4582 | 0.9911 |
| 1 | 0.01 | 1.0167 | 0.5120 | 0.5296 | 0.4670 | 0.3434 | **0.3546** | 0.9943 |
| 1 | 0 | 1.0697 | 0.4583 | 0.5269 | **0.37041** | 0.9976 | 0.4241 | 0.9877 |
| 0.1 | 1 | 0.9984 | 0.4546 | 0.4630 | **0.4382** | 0.9985 | 0.4801 | 0.6293 |
| 0.1 | 0.1 | 0.6707 | 0.5593 | **0.4868** | 0.4948 | 0.4887 | 0.5106 | 0.6486 |
| 0.1 | 0.01 | 0.5000 | 0.4893 | 0.4987 | 0.4520 | 0.2123 | **0.4445** | 0.7546 |
| 0.1 | 0 | 0.6251 | 0.5839 | **0.3859** | 0.57406 | 0.3009 | 0.4343 | 0.6805 |
| 0.01 | 1 | 1.3170 | 0.5309 | 0.5845 | 0.5454 | 0.9975 | **0.4127** | 0.4304 |
| 0.01 | 0.1 | 1.0503 | 0.5076 | 0.4541 | **0.4493** | 0.2746 | 0.4703 | 0.3426 |
| 0.01 | 0.01 | 0.6370 | **0.4638** | 0.4794 | 0.4858 | 0.1871 | 0.5633 | 0.2787 |
| 0.01 | 0 | 0.5665 | 0.5247 | 0.4977 | 0.4882 | 0.1947 | **0.4566** | 0.2326 |

Table 1: Results for Friedman3 dataset with linear model prior. The MAEs of cGAN, cGAN with TabNet generator and baseline linear model (Stats model) are 0.4477, 0.49724 and 0.5529 respectively.

weights are also reported. Tables 1, 4, 7 correspond to Friedman3 dataset, 2, 5, 8 to Boston dataset and 3, 6, 8 to Energy dataset. For tables 1, 2, 3, we use Linear Models as the prior, GBT in Tables 4, 5, 6 and TabNet in Tables 7, 8, 9. By looking at all the Tables I-IX collectively, it is clear that ABC-GAN models are able to detect the extent of misspecification, as the reduction in the MAE, relative to the prior model, is more pronounced for larger misspecifications. Hence we see that as the misspecification of the pre-generator increases, the model relies more and more on the GAN generator to do the correction. Overall, we notice that our model majorly outperforms SOTA models such as C-GAN, C-GAN with TabNet generator, TabNet regressor and CatBoost.

A skip connection has been added in some models, as explained earlier, to take a weighted average of the prior model and the GAN model. The weight given to the GAN in the skip connection tends to increase with increase in variance and bias, and is ideally supposed to be close to 1 for the highest variance and bias values and close to 0 for lowest variance and bias values. In most cases variance seems to be playing a greater role in the skip connection weight than the bias. This indicates that as the model misspecification increases, more weightage is given to the GAN skip node to help cofrrect this misspecification. Hence, as the complexity of the prior increases (such as when we use Transformers as priors), mGAN is sufficient to correct the misspecification of the models. However, for models with lower complexity (such as linear models), skipGAN performs better in correcting the model misspecification. From tables 4 and 6, it is evident that as the misspecification reduces, the skipGAN weight reduces and drops to almost 0 (it becomes 0 for Tab-skipGAN for variance 0.001 and bias 0). This effectively proves our original claim that when $G_\pi$ is almost perfect, $G_\gamma$ is almost an identity transformation and $d(D_\tau, D_\pi) \approx 0$. As the noise increases, the dependence on the GAN generator increases, resulting in higher weights in the skipGAN.

Using TabNet network for the generator of the GAN helps in stabilising the model. mGAN, Tab-mGAN and Tab-skipGAN perform consistently well with no high MAE outlier. While Tab-mGAN and Tab-skipGAN may not consistently outperform their vanilla counterparts (mGAN and skipGAN), adding the TabNet Network ensures consistent results across multiple iterations.

We also wanted to explore the effect of different sizes. We consider the Boston dataset again, and took a subset of the data to see if, as the dataset size increases, ABC-GANs continue to do well. As expected, the performance of the all the models improves with increase in sample size (as visible from Fig. 4 to Fig. 8). However skipGAN destabilizes for larger datasets (see tables 5, 6 and 8), thus resulting in large MAE values for a few experimental set-ups.

# 6  Discussion and Conclusion

Out of the numerous types of GANs available, how is ABC-GAN different? No other GAN works on the idea of correcting model misspecification. In this proposal, we specify that our emphasis was not on obtaining better performance, but it is to show that the model is capable of doing a likelihood-free inference, and

| Variance | Bias | Prior model | mGAN | Tab-mGAN | skipGAN | Weights skipGAN | Tab-skipGAN | Weights Tab-skipGAN |
|---|---|---|---|---|---|---|---|---|
| 1 | 1 | 1.2763 | 0.3017 | 0.3050 | 0.2876 | 0.9904 | **0.2818** | 0.9935 |
| 1 | 0.1 | 0.8442 | 0.2653 | 0.2876 | **0.2355** | 0.9934 | 0.2632 | 1.0000 |
| 1 | 0.01 | 0.9313 | 0.2975 | 0.2880 | **0.2684** | 0.9876 | 0.2978 | 0.9855 |
| 1 | 0 | 0.9459 | 0.2717 | 0.2831 | **0.2570** | 0.9963 | 0.2844 | 0.9930 |
| 0.1 | 1 | 1.0254 | 0.2991 | 0.3114 | **0.2986** | 0.7022 | 0.3188 | 0.8000 |
| 0.1 | 0.1 | 0.4154 | 0.3076 | 0.2587 | **0.2508** | 0.6869 | 0.3011 | 0.7775 |
| 0.1 | 0.01 | 0.3820 | 0.2968 | 0.2820 | **0.2790** | 0.7133 | 0.3057 | 0.7673 |
| 0.1 | 0 | 0.3779 | 0.2796 | 0.2761 | **0.2738** | 0.6000 | 0.2767 | 0.7441 |
| 0.01 | 1 | 1.0492 | 0.2559 | 0.2700 | **0.2530** | 0.3580 | 0.2568 | 0.3814 |
| 0.01 | 0.1 | 0.4038 | 0.2627 | 0.2573 | **0.2488** | 0.1698 | 0.2552 | 0.3393 |
| 0.01 | 0.01 | 0.3733 | 0.2847 | **0.2684** | 0.2785 | 0.1631 | 0.2888 | 0.4281 |
| 0.01 | 0 | 0.3712 | 0.3073 | 0.2899 | 0.3450 | 0.1592 | **0.2625** | 0.2849 |

Table 2: Results for Boston dataset with linear model prior. The MAEs of cGAN, cGAN with TabNet generator and baseline linear Model are 0.2838, 0.2729 and 0.3508 respectively.

| Variance | Bias | Prior model | mGAN | Tab-mGAN | skipGAN | Weights skipGAN | Tab-skipGAN | Weights Tab-skipGAN |
|---|---|---|---|---|---|---|---|---|
| 1 | 1 | 1.0849 | 0.1090 | 0.1098 | **0.1014** | 0.9973 | 0.1486 | 0.9991 |
| 1 | 0.1 | 0.8286 | **0.0752** | 0.1221 | 0.1058 | 0.9958 | 0.2130 | 0.9932 |
| 1 | 0.01 | 0.8344 | 0.1462 | 0.1557 | 0.1391 | 1.0000 | **0.0654** | 1.0000 |
| 1 | 0 | 0.8764 | 0.1461 | 0.1347 | **0.0943** | 0.9833 | 0.1209 | 0.9936 |
| 0.1 | 1 | 1.0036 | 0.1544 | 0.1089 | **0.0977** | 0.5256 | 0.1076 | 0.9568 |
| 0.1 | 0.1 | 0.2493 | 0.0955 | 0.0946 | 0.0916 | 0.3530 | **0.0630** | 0.9523 |
| 0.1 | 0.01 | 0.2184 | 0.1608 | **0.0538** | 0.0566 | 0.3431 | 0.0945 | 0.9520 |
| 0.1 | 0 | 0.2239 | 0.1352 | **0.0930** | 0.1315 | 0.3745 | 0.1455 | 0.9794 |
| 0.01 | 1 | 0.9740 | 0.0859 | **0.0794** | 0.1076 | 0.3152 | 0.0829 | 0.2947 |
| 0.01 | 0.1 | 0.2302 | 0.1677 | 0.0812 | **0.0762** | 0.2097 | 0.1354 | 0.2539 |
| 0.01 | 0.01 | 0.2246 | 0.2143 | 0.2038 | **0.1883** | 0.0646 | 0.2278 | 0.1681 |
| 0.01 | 0 | 0.2035 | 0.1274 | **0.0962** | 0.1169 | 0.1025 | 0.1576 | 0.2352 |

Table 3: Results for Energy efficiency dataset for 1st target with Linear model prior. The MAEs of cGAN, cGAN with TabNet generator and baseline Linear Model (stats model) are 0.0849, 0.1564 and 0.2008 respectively.

| Variance | Bias | Prior model | mGAN | Tab-mGAN | skipGAN | Weights skipGAN | Tab-skipGAN | Weights Tab-skipGAN |
|---|---|---|---|---|---|---|---|---|
| 1 | 1 | 1.0837 | **0.3539** | 0.5377 | 0.3579 | 0.5313 | 0.4726 | 0.5923 |
| 1 | 0.1 | 0.9548 | **0.3912** | 0.4745 | 0.4254 | 0.4767 | 0.4238 | 0.8411 |
| 1 | 0.01 | 0.9633 | **0.4096** | 0.4841 | 0.4250 | 0.4429 | 0.5243 | 0.7531 |
| 1 | 0 | 0.9922 | **0.4766** | 0.5013 | 0.5103 | 0.3814 | 0.5285 | 0.7223 |
| 0.1 | 1 | 1.1230 | 0.4252 | 0.4411 | **0.4232** | 0.5064 | 0.4284 | 0.2557 |
| 0.1 | 0.1 | 0.5224 | 0.5252 | 0.6032 | 0.5225 | 0.3797 | **0.5197** | 0.0671 |
| 0.1 | 0.01 | 0.3800 | 0.3560 | 0.4523 | 0.3902 | 0.1867 | **0.3805** | 0.0342 |
| 0.1 | 0 | 0.4564 | 0.4668 | 0.4601 | 0.4567 | 0.2389 | **0.4403** | 0.0534 |
| 0.01 | 1 | 1.0652 | 0.2989 | 0.5498 | **0.2512** | 0.0698 | 0.2735 | 0.1933 |
| 0.01 | 0.1 | 0.4599 | 0.4432 | **0.3931** | 0.4498 | 0.2812 | 0.4450 | 0.0464 |
| 0.01 | 0.01 | 0.4025 | 0.4027 | 0.4781 | 0.3989 | 0.0131 | **0.3982** | 0.0425 |
| 0.01 | 0 | 0.3792 | 0.3908 | 0.4349 | 0.3865 | 0.4759 | **0.3772** | 0.0000 |

Table 4: Results for Friedman3 dataset with Gradient Boosted Trees (GBT) prior. The MAEs of cGAN, cGAN with TabNet generator and baseline GBT (Catboost) Model are 0.4477, 0.49724 and 0.4215 respectively.

| Variance | Bias | Prior model | mGAN | Tab-mGAN | skipGAN | Weights skipGAN | Tab-skipGAN | Weights Tab-skipGAN |
|---|---|---|---|---|---|---|---|---|
| 1 | 1 | 1.1928 | 0.3492 | **0.2812** | 0.2888 | 0.9645 | 0.2834 | 0.8963 |
| 1 | 0.1 | 0.8688 | 0.2892 | 0.2765 | 0.2949 | 0.9308 | **0.2731** | 0.8692 |
| 1 | 0.01 | 0.8406 | 0.3559 | 0.2787 | 0.2493 | 0.9953 | **0.2732** | 0.8812 |
| 1 | 0 | 0.8120 | 0.2821 | **0.2447** | 0.2704 | 0.9742 | 0.2696 | 0.93485 |
| 0.1 | 1 | 1.0098 | 0.2704 | **0.2068** | 0.2662 | 0.1860 | 0.2420 | 0.2964 |
| 0.1 | 0.1 | 0.2419 | 0.3634 | **0.2189** | 0.2596 | 0.0382 | 0.2640 | 0.0917 |
| 0.1 | 0.01 | 0.2256 | 0.3437 | **0.2206** | 0.2306 | 0.0313 | 0.2271 | 0.0021 |
| 0.1 | 0 | 0.2700 | 0.4463 | **0.2646** | 0.3035 | 0.0297 | 0.2666 | 0.0209 |
| 0.01 | 1 | 1.0205 | 0.3271 | **0.2327** | 0.3547 | 0.1945 | 0.3541 | 0.2191 |
| 0.01 | 0.1 | 0.2562 | 0.3181 | 0.2551 | **0.2422** | 0.0317 | 0.2493 | 0.0599 |
| 0.01 | 0.01 | 0.2424 | 0.3075 | 0.2640 | 21.9047 | 0.1488 | **0.2166** | 0.0550 |
| 0.01 | 0 | 0.2198 | 0.2683 | 0.2287 | 0.2291 | 0.0058 | **0.2179** | 0.0361 |

Table 5: Results for Boston dataset with Gradient Boosted Trees (GBT) prior. The MAEs of cGAN, cGAN with TabNet generator and baseline GBT (Catboost) are 0.2838, 0.2729 and 0.2049 respectively.

| Variance | Bias | Prior model | mGAN | Tab-mGAN | skipGAN | Weights skipGAN | Tab-skipGAN | Weights Tab-skipGAN |
|---|---|---|---|---|---|---|---|---|
| 1 | 1 | 1.1591 | 0.1462 | **0.0902** | 0.1149 | 0.9594 | 0.0983 | 0.9882 |
| 1 | 0.1 | 0.7870 | 0.0915 | 0.0981 | 0.1154 | 0.9798 | **0.0733** | 0.9815 |
| 1 | 0.01 | 0.7771 | **0.0848** | 0.1636 | 0.1339 | 0.9432 | 0.1112 | 0.9927 |
| 1 | 0 | 0.8482 | 0.0924 | **0.0577** | 0.1334 | 0.9834 | 0.1549 | 0.9950 |
| 0.1 | 1 | 1.0073 | **0.0745** | 0.0762 | 0.0937 | 0.2482 | 0.0776 | 0.3692 |
| 0.1 | 0.1 | 0.1178 | **0.0783** | 0.1382 | 0.1077 | 0.0671 | 0.0906 | 0.1422 |
| 0.1 | 0.01 | 0.0787 | **0.0656** | 0.0750 | 0.1138 | 0.0979 | 0.0964 | 0.2580 |
| 0.1 | 0 | 0.0801 | **0.0637** | 0.0650 | 0.0830 | 0.0028 | 0.0823 | 0.0232 |
| 0.01 | 1 | 0.9994 | 0.0662 | 0.0762 | 0.1157 | 0.2280 | **0.0522** | 0.2164 |
| 0.01 | 0.1 | 0.1004 | 0.1231 | **0.0482** | 0.0489 | 0.0751 | 0.0698 | 0.0782 |
| 0.01 | 0.01 | 0.0248 | 0.0265 | 0.0436 | 1048.8400 | 0.0184 | **0.0233** | 0.0000 |
| 0.01 | 0 | **0.0249** | 0.1845 | 0.0650 | 0.0287 | 0.0333 | 0.0284 | 0.0034 |

Table 6: Results for Energy Efficiency dataset for 1st target with Gradient Boosted Trees (GBT) prior. The MAEs of cGAN, cGAN with TabNet generator and baseline GBT (Catboost) are 0.0849, 0.1564 and 0.0201 respectively.

| Variance | Bias | Prior model | mGAN | Tab-mGAN | skipGAN | Weights skipGAN | Tab-skipGAN | Weights Tab-skipGAN |
|---|---|---|---|---|---|---|---|---|
| 1 | 1 | 1.1267 | **0.4191** | 0.5205 | 0.4517 | 0.1495 | 0.5559 | 0.5722 |
| 1 | 0.1 | 1.0068 | **0.3514** | 0.4891 | 0.4270 | 0.1977 | 0.5288 | 0.7900 |
| 1 | 0.01 | 0.8485 | **0.3857** | 0.5234 | 0.4168 | 0.1942 | 0.5164 | 0.6472 |
| 1 | 0 | 0.9358 | **0.4052** | 0.4305 | 0.4102 | 0.2730 | 0.4468 | 0.7080 |
| 0.1 | 1 | 1.0669 | 0.4593 | 0.6085 | **0.4591** | 0.1444 | 0.5137 | 0.1770 |
| 0.1 | 0.1 | 0.3809 | 0.4044 | 0.4932 | 0.4477 | 0.1356 | **0.3329** | 0.0520 |
| 0.1 | 0.01 | 0.5561 | **0.4130** | 0.5409 | 0.4294 | 0.2035 | 0.5938 | 0.0309 |
| 0.1 | 0 | 0.4094 | **0.3674** | 0.4049 | 0.3981 | 0.0000 | 0.3828 | 0.0808 |
| 0.01 | 1 | 1.0446 | 0.3951 | 0.4414 | **0.3740** | 0.3830 | 0.4562 | 0.1855 |
| 0.01 | 0.1 | 0.4847 | 0.4940 | 0.5045 | **0.4416** | 0.1651 | 0.5273 | 0.1612 |
| 0.01 | 0.01 | 0.4274 | **0.4153** | 0.5523 | 0.5022 | 0.2027 | 0.5107 | 0.1883 |
| 0.01 | 0 | 0.4536 | **0.4328** | 0.4709 | 0.4409 | 0.0000 | 0.4727 | 0.1730 |

Table 7: Results for Friedman3 dataset with Trasformer network prior. The MAEs of cGAN, cGAN with TabNet generator and baseline transformer (TabNet) model are 0.4477, 0.49724 and 0.5529 respectively.

| Variance | Bias | Prior model | mGAN | Tab-mGAN | skipGAN | Weights skipGAN | Tab-skipGAN | Weights Tab-skipGAN |
|---|---|---|---|---|---|---|---|---|
| 1 | 1 | 1.2568 | 0.3098 | **0.2527** | 0.2791 | 0.9812 | 0.2560 | 0.9347 |
| 1 | 0.1 | 0.7969 | 0.2417 | 0.2544 | 0.2767 | 0.9747 | **0.2336** | 0.9272 |
| 1 | 0.01 | 0.7705 | 0.2943 | 0.2891 | 0.2719 | 0.9670 | **0.2385** | 0.9724 |
| 1 | 0 | 1.0423 | 0.2798 | **0.2754** | 0.3089 | 0.9857 | 0.2915 | 0.8586 |
| 0.1 | 1 | 1.0238 | 0.3667 | **0.1848** | 0.4119 | 0.2430 | 0.2313 | 0.3206 |
| 0.1 | 0.1 | 0.2796 | 0.2951 | 0.2685 | **0.2400** | 0.1145 | 0.2506 | 0.1688 |
| 0.1 | 0.01 | 0.2864 | 0.3200 | 0.2885 | 0.3356 | 0.1382 | **0.2665** | 0.2107 |
| 0.1 | 0 | **0.2318** | 0.3291 | 0.2455 | 288.3585 | 0.0923 | 0.2643 | 0.2122 |
| 0.01 | 1 | 1.0746 | 0.4723 | 0.3016 | 0.2897 | 0.2173 | **0.2697** | 0.2799 |
| 0.01 | 0.1 | 0.2694 | 0.2977 | **0.2459** | 527.8482 | 0.0105 | 0.3008 | 0.1342 |
| 0.01 | 0.01 | 0.2240 | 0.2725 | **0.2142** | 0.2820 | 0.0735 | 0.2957 | 0.2292 |
| 0.01 | 0 | 0.2089 | 0.2849 | **0.1980** | 2303.4068 | 0.1503 | 0.2628 | 0.3504 |

Table 8: Results for Boston dataset with Transformer network prior. The MAEs of cGAN, cGAN with TabNet generator and baseline transformer (TabNet) model are 0.2838, 0.2729 and 0.2515 respectively.

| Variance | Bias | Prior model | mGAN | Tab-mGAN | skipGAN | Weights skipGAN | Tab-skipGAN | Weights Tab-skipGAN |
|---|---|---|---|---|---|---|---|---|
| 1 | 1 | 1.2390 | 0.0618 | 0.0903 | **0.0499** | 0.9200 | 0.1203 | 0.9702 |
| 1 | 0.1 | 0.7765 | 0.1274 | 0.1342 | **0.0597** | 0.9715 | 0.0693 | 0.9976 |
| 1 | 0.01 | 0.7958 | 0.1380 | 0.0778 | 0.1183 | 0.9353 | **0.0708** | 1.0000 |
| 1 | 0 | 0.7588 | **0.0548** | 0.1465 | 0.0604 | 0.9753 | 0.0970 | 0.9916 |
| 0.1 | 1 | 1.0056 | 0.3290 | 0.1076 | **0.0551** | 0.3679 | 0.0640 | 0.4676 |
| 0.1 | 0.1 | 0.1517 | **0.0489** | 0.0511 | 0.1028 | 0.0894 | 0.0945 | 0.4901 |
| 0.1 | 0.01 | 0.1052 | 0.0974 | 0.0894 | **0.0689** | 0.1474 | 0.0998 | 0.0420 |
| 0.1 | 0 | 0.0907 | 0.0708 | **0.0651** | 0.1349 | 0.0665 | 0.0652 | 0.1287 |
| 0.01 | 1 | 0.9865 | 0.0987 | **0.0524** | 0.3475 | 0.3747 | 0.0897 | 0.2481 |
| 0.01 | 0.1 | 0.0942 | 0.1016 | 0.0872 | **0.0789** | 0.1056 | 0.0941 | 0.0000 |
| 0.01 | 0.01 | 0.0514 | 0.1431 | 0.0698 | 0.1271 | 0.0193 | **0.0477** | 0.0379 |
| 0.01 | 0 | 0.0652 | **0.0429** | 0.0840 | 0.1034 | 0.0591 | 0.1034 | 0.0451 |

Table 9: Results for Energy Efficiency dataset for 1st target with Transformer network prior. The MAEs of cGAN, cGAN with TabNet generator and baseline transformer (TabNet) model are 0.0849, 0.1564 and 0.0543 respectively.

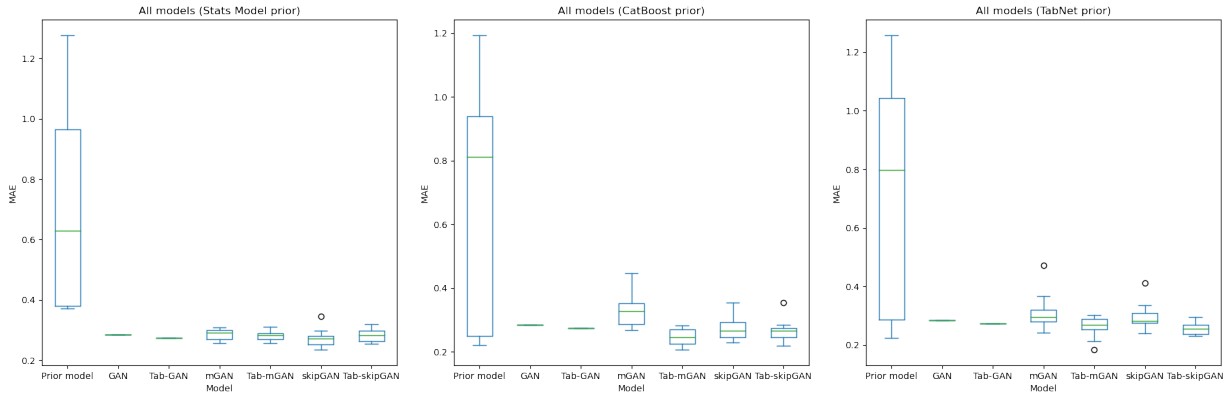

Figure 3: Box plots for comparison of models in the Boston dataset. All ABC-GAN models outperform the Linear, GBT and transformer prior models. Large outliers (MAE ≥ 20) for skipGAN were removed.
.

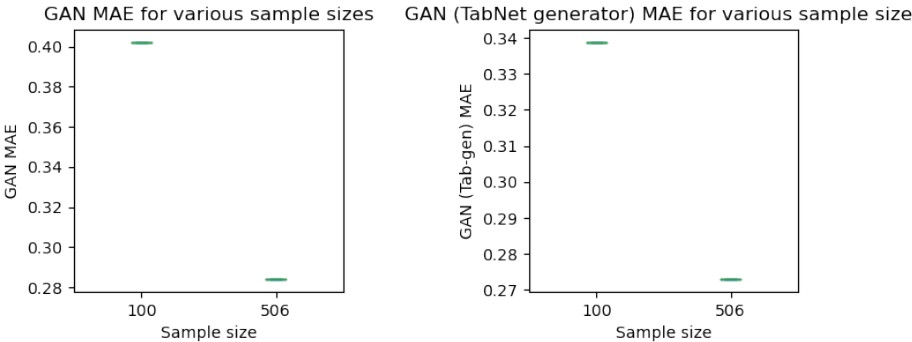

Figure 4: cGAN and cGAN with TabNet generator models on 100 samples and 503 samples (entire dataset) of Boston dataset.

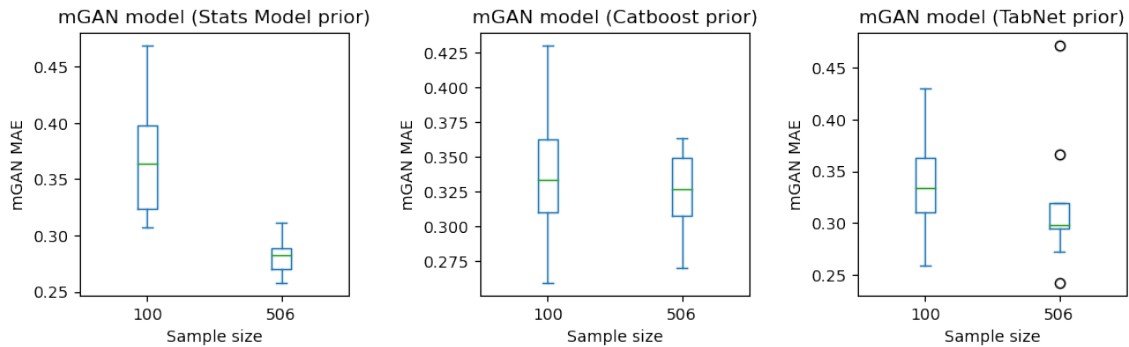

Figure 5: mGAN model for Linear model, GBT and Transformer priors on 100 samples and 503 samples (entire dataset) of Boston dataset.

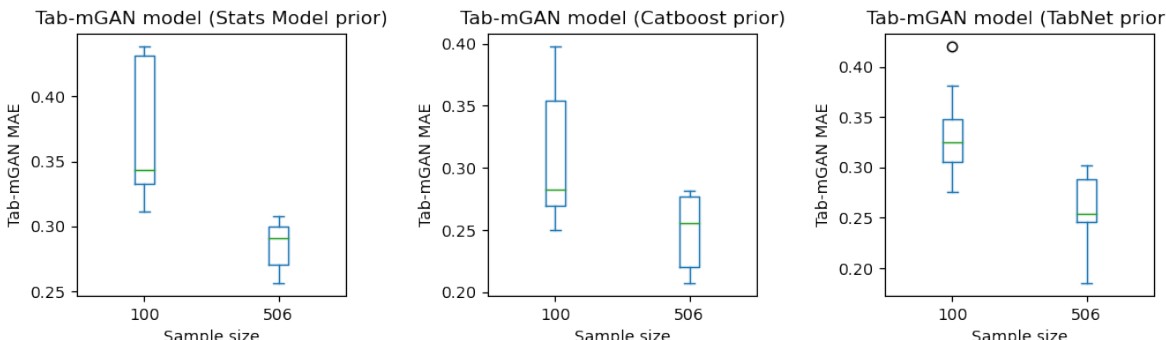

Figure 6: Tab-mGAN model for Linear model, GBT and Transformer priors on 100 samples and 503 samples (entire dataset) of Boston dataset.

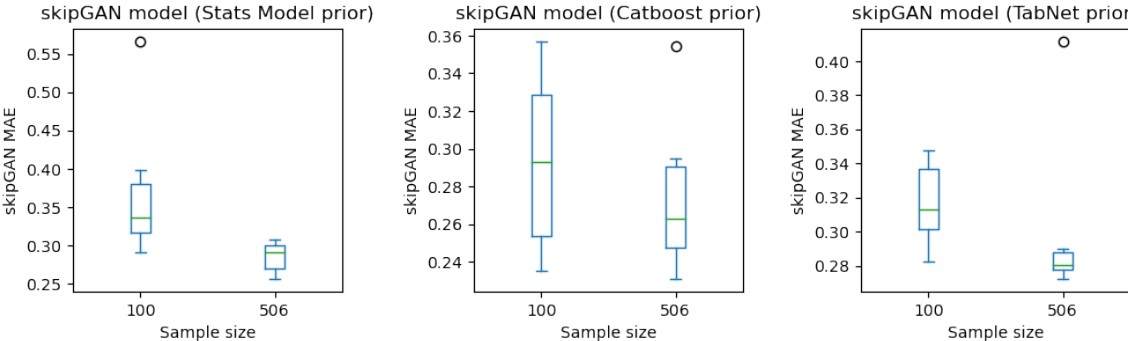

Figure 7: skipGAN model for Linear model, GBT and Transformer priors on 100 samples and 503 samples (entire dataset) of Boston dataset.

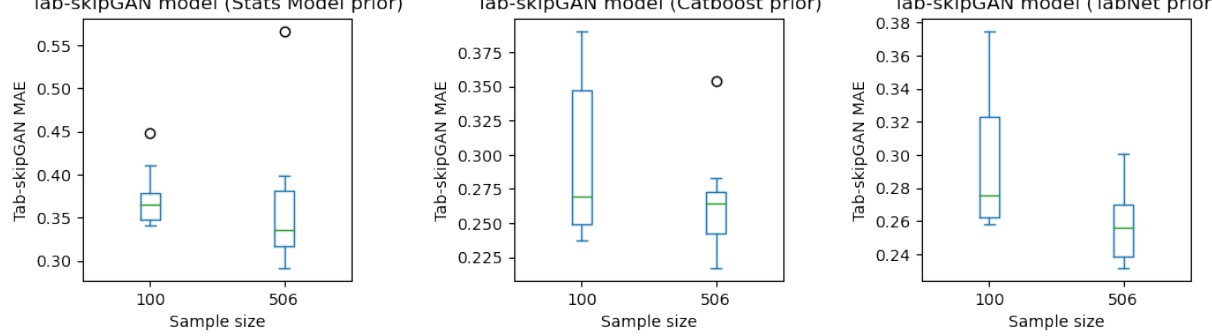

Figure 8: Tab-skipGAN model for Linear model, GBT and Transformer priors on 100 samples and 503 samples (entire dataset) of Boston dataset.

is more explicit in it's way of working than most pre-existing black-box models. Our ABC-GAN models outperform prior models with the same amount of misspecification, and perform equivalent or better than these priors even in the ideal situation of perfectly specified models.

How is our experimentation on regression any different than the other existing work, among the wide variety of literature that exists on regression, including non-parametric approaches such as Gaussian process regression [Wang]? While being useful in the ML community, these methods don't solve the problems of (1) correcting likelihood misspecification in the models or data and (2) performing equivalent or better than to the prior models under perfect condition (no noise condition). Our model caters mainly to correcting misspecification in the prior models, and performs equivalently or better than the prior models in the ideal case in several regression tasks.

In this paper, the objective that we want to achieve is to regularise the GAN generator by prepending the complex sampler(s), which ideally would have all the domain knowledge (which would be otherwise captured by the prior on the parameters in case of the ABC, thereby biasing the training of the GAN). In this case, although there is just one complex sampler initially, we have multiple samplers - one for each candidate model, with each sampler trying to learn a different transformation. The distance between the simulated and actual data is measured using a divergence metric and ultimately only those samplers or models are chosen which lie within a certain threshold. We argued that, the proposed method can do no worse than the baselines, but also significantly outperforms the baseline priors, and can successfully correct the likelihood misspecification in them [1]. Hence, in the ABC-GAN framework, the Generator is correcting for the misspecification, while the Discriminator is learning summary statistics (data representations) along with the rejection region. Our simple and elegant formulation can absorb a variety of paradigms. It will be interesting to investigate a full Bayesian setup, and draw posterior samples for the baseline. Likewise, on the adversarial optimization side, owing to incorporation of prior knowledge, stability dynamics could be studied. Our extensive experimentation involving wide variety of datasets, baseline models and tasks reaffirms our belief that, the proposed regime can be used for continuous model improvement, in an inter-operable way.

Our work opens up many future directions. In our current work, we have not yet exploited obtaining posterior inference. Can we compute the posterior quantities, like in ABC? A reasonable hunch is to calculate approximate posterior quantities, under change of measure. Here, we view $T \equiv G_{GAN}$ as fixed, deterministic, but differential transformation. Recent advances in gradient-based normalizing flows inspire us in this direction [Song & Ermon (2019)]. It is relatively straightforward to obtaining posterior predictive distribution - sample from the ABC-Pre Generator, and pass it through GAN Generator, and treat the samples as approximate draws with which any statistics can be computed. Another interesting question to ask is - does the discriminating function learnt by $D_{GAN}$ approximate the Bayes Factor and/or the likelihood ratio? Previous work in this direction provide hints [Shirazi et al. (2017)]. Likewise, it will be of interest to know whether the representations learnt by $D_{GAN}$ showcase sufficient statistics? Earlier work on learning summary statistics via deep neural networks for ABC provide clues [Wong et al. (2018)]. Under linear models or generalized linear models, we find an affirmative answer. From stability standpoint, can the specific type of regularization of $G_{GAN}$ be tuned such that an optimum between explicit and implicit generative models is found? Pursuing the above questions will help us understand ABC-GANs better.

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
