# OpenReview forum: "Correcting Model misspecification via Generative Adversarial Networks"
_TMLR — Rejected by TMLR_

### Review · Reviewer_uyLU · 2023-10-23

**Summary Of Contributions:**

Given data $y_i^\tau$ and samples from a misspecified generative model $y_i^\pi$, the authors propose to use a GAN to make $y_i^\tau$ and $y_i^\pi$ indistinguishable. The resulting GAN has learned to correct for the model misspecification. The authors demonstrate the efficacy of their approach when the model misspecification is additive Gaussian noise.

**Audience:**

Yes

**Claims And Evidence:**

No

**Requested Changes:**

* Add a rigorous explanation on how ABC models are trained.
* Add a rigorous explanation on how GANs are trained.
* Define model misspecification precisely, and explain how it can occur in practice, and what negative effects it has.
* Include an experiment where the model misspecification occurs naturally.

**Strengths And Weaknesses:**

The authors tackle an interesting question, which is to correct for the mismatch between the data generating process and model.

This paper was difficult to understand because of the lack of relevant background information. Approximate Bayesian Computation (ABC) is a major part of this paper, but there is almost no explanation on what it is, and how such models are trained. The paper would also benefit from a brief, yet rigorous explanation on how GANs are trained.

The authors' design choice of simulating model misspecification as additive Gaussian noise also seems too simplistic to be relevant. The only baseline method included is the prior model, and the fact that the GAN-based approach is performing better seems to just show that GANs can denoise additive Gaussian noise.

It would be more interesting to demonstrate how model misspecification can arise naturally, and that the proposed approach can mitigate it.

---

> ### Author Response · Authors · 2023-10-23
> **Response**
>
> We thank the reviewer for the comments. We will address each comment soon and revise the manuscript as suggested.

---

> ### Author Response · Authors · 2023-11-05
> **Response window**
>
> We thank the reviewers for the thoughtful reviews. As we prepare our response, we can't help but notice that that we 're expected to respond with all requested changes within two weeks. While preparing response is definitely possible within the given window, some reviewers have asked for additional experiments and results. We think time is too short for such experimentas. Therefore, we request an additional 2 weeks time (December 3). We hope that our request will be granted.
> Thanks again.

---

> ### Author Response · Authors · 2023-12-06
> **Introduction to ABC and GANs and other suggestions will be incorporated in the revised manuscript**
>
> **This paper was difficult to understand because of the lack of relevant background information. Approximate Bayesian Computation (ABC) is a major part of this paper, but there is almost no explanation on what it is, and how such models are trained. The paper would also benefit from a brief, yet rigorous explanation on how GANs are trained**
>
> **Requested Changes:
> Add a rigorous explanation on how ABC models are trained.
> Add a rigorous explanation on how GANs are trained.
> Define model misspecification precisely, and explain how it can occur in practice, and what negative effects it has.
> Include an experiment where the model misspecification occurs naturally.**
>
> Yes, it will be addressed in the revised manuscript.
>
> **It would be more interesting to demonstrate how model misspecification can arise naturally, and that the proposed approach can mitigate it**
>
> Yes, we will be adding it in the revised manuscript. For tabular data, it is a combination of feature engineering and model choice - both contribute to model misspecification, assuming that, there is sufficient is signal intrinsically in the data w.r.t the task at hand.

---

### Review · Reviewer_FyEW · 2023-11-04

**Summary Of Contributions:**

The paper studies the problem of generative modeling. The paper proposes to prepend the GAN model with the generating model of approximate Baysian computation so that GAN and ABC can benefit from each other. On the one hand, GAN can incorporate subject knowledge into the generation procedure and possibly use a smaller architecture. On the other hand, ABC does not need to require accurate information as GAN can correct the misspecification. The paper shows that the proposed ABC-GAN method can out-perform the baselines on some tasks.

**Audience:**

Yes

**Broader Impact Concerns:**

I happen to know that there are some ethical concerns related to using the Boston Housing dataset. You can check the sklearn page of the dataset for more information. Other than the usage of that dataset, I have no other ethical concerns.

**Claims And Evidence:**

Yes

**Requested Changes:**

1. Apply the proposed method to larger generation tasks, e.g., some image generations.

**Strengths And Weaknesses:**

Strengths:
1. The paper proposes a novel idea that combines ABC and GAN so that the two methods can benefit from each other.


Weaknesses:
1. The experimental settings all use datasets of relatively small scales. Whether such an approach can be applied to larger datasets, more complicated data distributions, and whether more complex GAN models can handle the misspecification remain to be answered.

2. The skip-based models do not always give good performance and sometimes drastically bad results. Ideally, the skip models should be as good as the non-skip models. It is not well understood why this is the case.

3. The results are presented using many large tables, which could be hard to parse. I would suggest the authors find better visualization methods to make the results more accessible.

4. As mentioned in the paper, it would be a better idea to accompany ABC with normalizing flows rather than GANs, as theoretical analysis would be more practical for using normalizing flows. Also, I don't see a clear advantage in why ABC has to be paired with GANs explicitly.


Minor:
1. The discriminator notation and the dataset notation are too similar.
2. "At the least, it should be able to ensure that the mGAN does at least as well as the baseline" redundant usage of "at least".
3. "Gradient boosting trees" and "gradient boosted trees" are both present in the paper. It's better to make the naming more consistent.
4. "The generative model for y is is nonlinear"
5. "and 3, 6, 8 to Energy dataset."

---

> ### Author Response · Authors · 2023-12-06
> **Tried another implementation of skip connection, which improves stability but does not lend the same interpretation.**
>
> **The skip-based models do not always give good performance and sometimes drastically bad results. Ideally, the skip models should be as good as the non-skip models. It is not well understood why this is the case**
>
> The motivation to introduce skip variants was precisely what the reviewer alluded to. In addition, we also consider this simple architectural modification to see if any interpretation can be offered. We have implemented the skip connection as a hard-constraint via clipping. As is evident, this choice of implementation retained the interpretability of the skip connection (when the model is correct, GAN Generator weight is close to 0, and is close to 1 in the other case). But this implementation choice has to lead to high variability in the model runs. To investigate this implementation choice, we considered sigmoid activation to ensure that, the weight is naturally constrained to be in the 0 to 1 range and initialized at 0.5. We ran the same experimental setup on Friedman3 dataset. Our observations revealed that, the runs were very stable, and this new skip model also consistently performed better (both skip mGAN and skip TabGAN) versions but the did not retain the interpretation we were looking for, i.e, the weight was never close to 0 or 1, when run for the same number of epochs. Overall, from a model selection standpoint, it may be preferable to use Transformer family as the default GAN architecture with and without skip connections and pick a model with better performance. This also inspires us to consider other simpler weighting schemes like a linear combination (not a convex combination) as other variants of the skip connection. It is to be noted that, our GAN implementation is very simple and it would be interesting to try GAN regularization techniques to see if the stability improves[Chu'2020, Kodali'2018, Salimans'2016].
>
>
> **The results are presented using many large tables, which could be hard to parse. I would suggest the authors find better visualization methods to make the results more accessible**
>
> Yes, tables will presented in visual form to read the data better.
>
>
> **As mentioned in the paper, it would be a better idea to accompany ABC with normalizing flows rather than GANs, as theoretical analysis would be more practical for using normalizing flows. Also, I don't see a clear advantage in why ABC has to be paired with GANs explicitly**
>
> As responded before, there are strong theoretical connections between ABC and GANs, and it would interesting to consider other generative paradigms like score models, diffusion models as an extension and continuation of the work.
>
>
> Other minor changes suggested will be incorporated in the revised manuscript.
>
> [Chu'2020] Chu, C., Minami, K., and Fukumizu, K., Smoothness and Stability in GANs, ArXiv abs/2002.04185 (2020)
>
> [Kodali'2018] Kodali, N., Hays, J., Abernethy, J., and Kira, Z., On Convergence and Stability of GANs, ArXiv abs/1705.07215 (2018)
>
> [Salimans'2016] Salimans, T., Goodfellow, I., Zaremba, W., Cheung, V., Radford, A., and Chen, X., Improved Techniques for Training GANs, ArXiv abs/1606.03498 (2016)

---

### Review · Reviewer_VFHR · 2023-11-04

**Summary Of Contributions:**

This work proposes to use a DNN model to transform the outputs of a simpler statistical model, and use a GAN-like objective to minimize an approximate divergence between the model and data distribution.  On several tabular regression datasets the method is demonstrated to achieve better performance compared to baseline methods.

**Audience:**

No

**Claims And Evidence:**

No

**Requested Changes:**

All issues mentioned above should be rectified.

**Strengths And Weaknesses:**

**Strengths.**
- Transforming the outputs of a simpler statistical model using a deep model is a sensible idea, although I am not sure about its originality.

**Weaknesses.**
- The manuscript does not appear to be using the terminology of ABC in a standard way.  ABC is generally an *inference* approach used to obtain a posterior distribution over the parameters of interest.  Fitting tree models and transformers on data (presumably doing point estimation) are not usually referred to as ABC.
- The manuscript claims to propose a new generative modeling paradigm, yet it actually describes a conditional modeling method which is only evaluated on univariate regression data. Furthermore, experiments do not evaluate the quality of the conditional distribution estimate, only reporting the MAE metric for regression.
- If only (conditional) medians are of interest, the MAE can be directly minimized; there is no need to introduce GAN discriminators.
- If full distributions are needed, it seems clear that invertible models such as normalizing flows should be preferred over GANs, as they make likelihood training possible and are no less flexible, especially on the low-dimensional tasks considered in the experiments. At the very least there should be experimental comparisons.

---

### Decision · Action_Editor_7J8P · 2023-12-28

**Recommendation:** Reject

**Comment:**

The aforementioned issues regarding soundness, experimental setting, and evidence were not cleared during the discussion phase. As such, I have to recommend rejection at this stage. Thank you for your submission, and I hope the remarks will help improve the paper.

**Audience:**

Yes.

**Claims And Evidence:**

Claims:

The paper introduces a generative framework that combines GAN with a simulator by first simulating the example and then correcting the misspecification (of the simulator) by training GAN on the concatenated input and simulation output. Adding the simulator as output is motivated from the perspective of added interpretability and regularization.

Evidence:

Reviewers challenged the notion that combining a GAN with a base model naturally leads to interpretability.

Reviewers were also not convinced by the experimental setting showing added benefit of the framework, in particular due to the small scale of the datasets used. The experimental setting does not compare to methods not using GANs, but still leveraging output of a simulator. This is a natural idea and as such it is expected that the literature contains other instantiations of the framework. If not, it should be more clearly argued by the Authors.

There were also some issues with refering to the method as ABC. The core to the ABC method is that it is a likelihood-free inference method, while in the case of the paper, it is used to refer to any generative model that takes as input output of a simulator.